# Parathyroid Hormone Concentration in Dogs Affected by Acute Kidney Injury Compared with Healthy and Chronic Kidney Disease

**DOI:** 10.3390/vetsci12020131

**Published:** 2025-02-06

**Authors:** Jari Zambarbieri, Erika Monari, Francesco Dondi, Pierangelo Moretti, Alessia Giordano, Paola Scarpa

**Affiliations:** 1Department of Veterinary Medicine and Animal Sciences, University of Milan, Via dell’Università 6, 26900 Lodi, Italy; jari.zambarbieri@guest.unimi.it (J.Z.); pierangelo.moretti@unimi.it (P.M.); alessia.giordano@unimi.it (A.G.); paola.scarpa@unimi.it (P.S.); 2Department of Veterinary Medical Sciences, Alma Mater Studiorum—University of Bologna, Via Tolara di Sopra n 50, 40064 Ozzano dell’Emilia, Italy; erika.monari3@unibo.it

**Keywords:** parathyroid hormone, hyperparathyroidism, dog, acute kidney injury, chronic kidney disease

## Abstract

This study aimed to investigate parathyroid hormone (PTH) status during canine acute kidney injury (AKI); while renal secondary hyperparathyroidism (RSHPT) is known to be a consequence of chronic kidney disease (CKD), there are no data regarding PTH concentrations in AKI patients, to date. Including three study groups, e.g., healthy patients, AKI patients and CKD dogs, respectively, biochemical parameters and PTH concentrations were compared between study groups. The PTH concentrations were significantly higher in both the AKI and CKD groups compared to healthy patients but without significant differences between these two latter study groups. In the AKI group, increased PTH concentrations were detected in 88.6% of dogs. Based on the results of this study, higher PTH concentrations also occurred in canine AKI, presumably as a result of the ionized hypocalcemia and hyperphosphatemia reported in these patients. On the other hand, PTH was not a useful tool for distinguishing AKI from CKD patients. The novelty of these data should be considered as a first step to a better understanding of PTH concentrations in canine AKI, knowing that future prospective studies are needed to better elucidate PTH kinetics over time in these patients.

## 1. Introduction

Acute kidney injury (AKI) is defined as a rapid decline in renal function leading to retention of uremic wastes, alterations in fluid balance and imbalances of electrolyte and acid-base status [1]. The abrupt decline of kidney function is recognized by an increase in serum creatinine (sCr) of ≥0.3 mg/dL within 48 h with a possible decrease in urine output. The mortality rate in dogs is high, approximately 50% [2,3,4].

Chronic kidney disease (CKD) is defined by the presence of structural or functional abnormalities of one or both kidneys, lasting for at least three months, regardless of the cause [5]. CKD impacts both the life expectancy and quality of life in affected dogs, depending on the severity of the disease.

AKI and CKD are currently classified using grading (from I to V) and staging (from 1 to 4) systems, respectively, based on sCr concentration, developed by the International Renal Interest Society (IRIS) [6].

Differentiating between AKI and CKD can be challenging in some cases. Both conditions are characterized by alterations in kidney functional markers, such as sCr, which reflect a decrease in kidney function in both clinical scenarios, although the temporal pattern of these changes and the underlying pathophysiology differ significantly.

Serum creatinine is insensitive to acute changes in renal function, and its levels vary with age, muscle mass, diet, medications and hydration status. Furthermore, it is not a direct marker of tubular damage but rather an indirect marker of GFR, which can increase also in cases of renal hypoperfusion, even when the kidneys are structurally intact (pre-renal azotemia) [7]. For these reasons, sCr is considered an ‘imperfect gold standard’ for the diagnosis of AKI [8].

Another issue with sCr is that, in most clinical situations, the patient’s true baseline value is not known, making the evaluation difficult [7].

Differentiation between acute and chronic renal disease is primarily based on clinical history, physical examination, clinicopathological data and ultrasonographic findings [9]. Reusch et al. (2000) proposed the ultrasonographic measurement of parathyroid gland size as a tool for differentiating acute from chronic renal failure. Parathyroid glands were significantly larger in dogs affected by chronic disease if compared to dogs with acute disease, although the study only included dogs with severe azotemia [10]. In any case, parathyroid gland size appears to be correlated with body weight, and the measurement procedure is not always feasible in clinical practice and is no longer recommended [10]. Other studies have proposed alternative biomarkers, such as plasma and urinary Neutrophil Gelatinase-Associated Lipocalin (NGAL), for differentiating AKI from CKD. However, this approach has several limitations, including sensitivity, specificity and the availability of laboratory methods for measurement [11,12,13,14].

Renal secondary hyperparathyroidism (RSHPT) represents a common complication of CKD, detected in 76% of affected dogs with an increasing prevalence related to disease severity, as reported in a previous study [15]. The increase of serum PTH concentration is caused by a derangement of the mineral metabolism due to a complex pathophysiological mechanism involving phosphate, the initial trigger, Fibroblast Growth Factor-23 (FGF-23), vitamin D metabolites and ionized calcium (iCa) [16,17].

In human medicine, it is demonstrated that some of the mineral metabolites’ abnormalities detected in CKD are also common during AKI: hypocalcemia, hyperparathyroidism (HPT), hyperphosphatemia, decreased vitamin D, increased FGF-23 levels and decreased renal Klotho expression [18]. Specifically, in human AKI, HPT is mainly driven by hypocalcemia, directly regulating PTH production by the parathyroid glands through a feedback mechanism, and secondarily promoted by hyperphosphatemia, which can chelate calcium, and by decreased calcitriol concentrations [18,19].

The clinical relevance of HPT in AKI has not yet been clarified, as higher PTH concentrations are not correlated to an increased risk of developing AKI in critically ill human patients nor correlated with an increased risk of 60-day mortality in patients with established diagnosed AKI and in critically ill patients [20,21]. In addition, two clinical studies and other less relevant research suggested that the magnitude of the increase in PTH concentration, with or without data about the kidney ultrasonographic aspect, may help distinguish between AKI and CKD. A sensitivity and specificity of 89%, respectively, were achieved using a PTH cutoff concentration of 170 pg/mL [22,23,24].

In veterinary medicine, similar clinicopathological abnormalities are observed in both canine AKI and CKD, including increased serum creatinine, urea and phosphate concentrations, as well as decreased blood pH, bicarbonate and urine specific gravity [2,25]. However, data about PTH concentrations in dogs with AKI are lacking, and the potential use of PTH as a variable to differentiate AKI from CKD remains unknown.

Therefore, hypothesizing higher PTH concentrations during canine AKI, as in human medicine, the aim of the study was to investigate PTH concentration status in AKI patients and to compare biochemical parameters and PTH concentrations among healthy dogs, dogs with AKI and CKD patients.

## 2. Materials and Methods

This retrospective observational study included dogs affected by AKI or CKD and a group of healthy dogs defined as controls. The patients were evaluated at the Veterinary Teaching Hospital of the Universities of Milan and Bologna during routine clinical activity between January 2019 and December 2021.

All the dogs underwent a physical examination following standard veterinary procedures, and blood samples were collected for diagnostic purposes. The biological samples were obtained with the owner’s consent. According to the Ethics Committee of the University of Milan (EC decision 29 October 2012, renewed with the protocol n° 02-2016), biological samples collected in these settings could also be used for research purposes without requiring a formal ethical approval.

The healthy group consisted of dogs referred for periodic check-up or for evaluation before elective surgery, using the following exclusion criteria: age under 1 year, pregnancy, history or clinical signs indicating disease within the previous 3 months, use of medications (except antiparasitic drugs) within the last 3 months, hematologic and biochemical abnormalities, presence of proteinuria (UPC ≥ 0.5) and inadequate urine specific gravity (USG < 1.030) [6].

CKD diagnosis was based on signalment, history, clinical signs and, in accordance with IRIS guidelines, using at least one of the following criteria: persistent sCr levels above 1.4 mg/dL, persistent renal proteinuria (UPC ratio > 0.5) or kidney ultrasonographic abnormalities suggestive of CKD. AKI was diagnosed according to IRIS guidelines based on signalment, history, clinical signs, sCr concentration, abdominal ultrasound and urine output evaluation [26]. All the dogs affected by kidney disease enrolled in the study were staged or graded according to the IRIS staging system for CKD and the grading system for AKI [6]. AKI grading was performed upon admission, specifically at the time of AKI diagnosis. Two to five mL of blood were collected at admission from the cephalic vein and placed into methacrylate tubes without anticoagulants, pre-filled with a gel separator and clot activator (FL Medical, Torreglia, Italy).

Biochemical analyses for diagnostic purposes and staging were performed within 2 h using automated analyzers (BT 3500, Biotecnica Instruments, Rome, Italy; AU480, Beckman Coulter, Brea, CA, USA) after the samples’ centrifugation (10 min, 2500× *g*) within 30 min from collection. An aliquot of 0.3 mL of whole blood was collected in electrolyte-balanced heparin syringes for venous blood gas analysis and iCa measurement, performed within 5 min of collection using the Stat Profile phOX Ultra Analyzer (Nova Biomedical, Lainate, Italy) and the ABL825 (Radiometer Medical APS, Brønshøj, Denmark). Leftover serum was frozen at −20 °C for subsequent PTH measurement (within 6 months from the collection) using an automated analyzer (AIA 360^®^) and a two-site immunoenzymatic assay (ST AIA-PACK^®^ Intact PTH, Tosoh Bioscience, Tessenderlo, Belgium), validated for use in dogs [27].

The PTH reference interval was determined using samples from the healthy group. An Excel spreadsheet (Microsoft Corp., Redmond, WA, USA) with the Reference Value Advisor (version 2.0) macro was used for the calculations [28]. The software performs computations according to the IFCC-CLSI recommendations as suggested by the ASVCP guidelines [29,30]. Descriptive statistics, a normality test (according to the Anderson–Darling method with histograms and Q–Q plots) and Box–Cox transformation were conducted. The Tukey test was used to detect outliers and suspected outliers. A robust method was applied for computation. Further statistical analysis was conducted using JMP 16 (SAS Inc., Cary, NC, USA) including descriptive analysis and the following tests: Anderson–Darling tests to investigate normality of data distribution and Kruskal–Wallis tests to compare PTH and other analyte concentrations among the AKI, CKD and healthy group, as well as between AKI grades. Spearman correlation tests were used to analyze the relationships between PTH concentration and sCr, phosphate (P), total calcium (tCa) and iCa. Dunn’s test was applied as a post-hoc analysis.

## 3. Results

### 3.1. Population

The healthy group was composed of 41 dogs. The median age was 6 years (range 1–14). Among these, 18 were females (12 neutered and 6 intact) and 23 were males (7 neutered and 16 intact). Of the 41 dogs, 14 were mixed-breed dogs, while 27 were purebred. The purebred dogs included 4 Dobermanns, 4 Labrador Retrievers, 2 Miniature Poodles and 2 Border Collies, with the remaining 15 breeds represented by 1 dog each.

The CKD group was composed of 35 dogs. The median age was 9 years (range 1–15). Of these, 20 were females (13 neutered and 7 intact) and 16 were males (4 neutered and 11 intact). Thirteen were mixed-breed dogs, while twenty-two were purebred dogs of various breeds: three Boxers, three Dobermanns and two Labrador Retrievers, with the remaining thirteen breeds represented by one dog each. According to the IRIS CKD staging system, the CKD dogs were distributed as follows: 7 in stage 1, 9 in stage 2, 9 in stage 3 and 10 in stage 4.

The AKI group was composed of 35 dogs. The median age was 8 years (range 1–15). Of these, 19 were females (13 neutered and 6 intact) and 16 were males (1 neutered and 15 intact). Twelve were mixed breed dogs, while twenty-three were purebred dogs of different breeds: three German Shepherds, two Fox Terriers and two Labrador Retrievers. The other 15 breeds were represented by 1 dog each. Considering the IRIS AKI grading system, the distribution of dogs included in the AKI group was as follows: 7 in grade I, 4 in grade II, 9 in grade III, 11 in grade IV and 4 in grade V. Considering the age in the different groups, a significant difference was found between the healthy and AKI group (*p* = 0.010).

### 3.2. PTH in Healthy Dogs

The PTH reference interval was determined and is graphically described below. A Gaussian distribution of data was shown after Box–Cox transformation (Figure 1) and by the Q–Q Plot (Figure 2). No outliers were detected.

In the healthy dog group, the PTH concentration ranged from 1.8 pg/mL to 14.7 pg/mL (mean ± SD: 6.92 pg/mL ± 2.93; median: 6.4 pg/mL; IRQ: 4.3).

The calculated lower limit was 2.0 pg/mL (CI 90% 1.5–2.7), while the upper limit was 13.8 pg/mL (CI 90% 11.9–14.7). The distribution of PTH concentrations is shown in Figure 3.

### 3.3. Biochemical Parameters in Healthy Dogs Versus AKI and CKD Dogs

The concentrations of selected biochemical parameters in the different groups are reported in Table 1.

Comparing biochemical variable results among the different groups, statistical analysis did not reveal significant differences in sCr concentrations between the AKI and CKD groups. Phosphate concentration differed significantly among the three groups, with higher concentrations observed in the AKI group compared to the CKD and healthy groups (*p* < 0.001) (Figure 4). No significant difference was detected in tCa among the three groups. Conversely, iCa was significantly lower in the AKI group (*p* = 0.002). Among the AKI dogs, 9 of 28 had an iCa concentration below the lower limit of the reference interval (LRL) of 1.2 mmol/L, with values ranging from 0.81 to 1.19 mmol/L and a median value of 1.21 mmol/L. Only one sample had iCa above the upper limit of the reference interval (URL). Total calcium concentration was below the LRL in 1 of 27 samples and above the URL in 3 samples.

In the CKD group, 4 out of 34 dogs had an iCa concentration below LRL, with values ranging from 0.83 to 1.19 mmol/L and a median value of 1.30 mmol/L. Ionized calcium was above the URL in three samples. The total calcium concentration was below the LRL in 4 out of 33 samples and above the URL in 6 samples.

### 3.4. PTH in AKI, CKD and Healthy Groups

No significant difference was detected in PTH concentrations between the AKI and CKD groups, while both groups were significantly different from the healthy group (*p* < 0.001) (Figure 5).

Considering the AKI group, only 4 of 35 (11.4%) dogs had PTH concentrations overlapping those observed in dogs without kidney disease. Among these, three dogs were classified as AKI grade I and one as AKI grade IV. Based on these data, PTH concentrations beyond the reference interval were identified in 31 of 35 AKI cases (88.6%).

In the CKD group, an increased concentration of PTH was observed in 24 of 35 (68.5%) dogs. Among the 11 dogs without increased PTH, 4 were staged in CKD stage 1, 6 in CKD stage 2 and 1 in CKD stage 3. All the dogs in CKD stage 4 had a PTH concentration above the reference interval.

The PTH concentrations in different AKI grades and CKD stages are reported in Table 2.

The distribution of PTH concentrations across AKI grades is shown in Figure 6. The Kruskal–Wallis test showed a significant difference among grades (*p* = 0.008). Post-hoc analysis confirmed a significant difference only between grade I and grade V (*p* = 0.013), despite the tendency for PTH concentrations to increase with the progression of the AKI grade.

In the AKI patients, according to Spearman test, PTH resulted as significantly correlated with sCr (*p* < 0.001; r = 0.67; CI: 0.604, 0.884), P (*p* < 0.001; r = 0.74; CI: 0.652, 0.913)) and iCa (*p* = 0.003; r = −0.52; CI: −0.706, −0.095), while it was not significantly correlated with tCa (*p* = 0.2).

In the CKD patients, PTH was significantly correlated with sCr (*p* < 0.001; r = 0.8; CI: 0.558, 0.868) and P (*p* < 0.001; r = 0.67; CI: 0.462, 0.835), while it was not significantly correlated with tCa and iCa (*p* = 0.200 and 0.080, respectively).

In healthy dogs, PTH was not significantly correlated with the parameters considered above.

## 4. Discussion

Renal secondary hyperparathyroidism (RSHPT) is a frequent consequence of CKD in humans and dogs [15,31]. In human medicine, increased concentrations of PTH have been frequently described in patients with AKI, although the clinical relevance of PTH in this condition remains unclear and does not appear to influence prognosis [18,20]. In veterinary medicine, however, data on the prevalence of elevated PTH concentration in dogs with AKI are lacking. This study shows that high concentrations of PTH are also a frequent finding in dogs with AKI, similar to what has been reported in canine CKD and human AKI. Additionally, the magnitude of PTH increase is associated with the severity of the disease, as represented by sCr concentrations and IRIS AKI grades. These results are consistent with those reported in CKD, both in canine and human patients [15,27,32].

Analyzing PTH concentrations, it is noteworthy that, as seen in CKD, during AKI, higher PTH concentrations can also be detected in dogs with normal sCr, likely as a quick homeostatic response to preserve calcium and phosphate balance.

Although PTH concentration did not significantly differ between the AKI and CKD groups, the underlying pathophysiological mechanisms likely differ between the two syndromes. In the study by Reusch et al., the size of parathyroid glands was significantly different when comparing dogs with severe azotemia due to AKI and those with CKD [10]. This difference is probably due to the fact that RSHPT in CKD is driven by parathyroid glands hyperplasia, while, in AKI, elevated PTH concentrations are probably due to the rapid release of preformed PTH stored in secretory granules and a swift increase in hormonal synthesis [33]. The duration of stimulation on the parathyroid glands, prolonged in CKD and shorter in AKI, may lead to anatomical differences that do not correspond to variations in laboratory findings, as PTH concentrations were elevated in both groups without significant differences. These findings suggest that PTH evaluation should not be used to distinguish AKI from CKD in the absence of a clear history of CKD or previous clinicopathologic data for comparison.

It is important also to evaluate whether the term hyperparathyroidism is appropriate in patients affected by AKI. In fact, sporadic cases of markedly elevated PTH have been reported in human patients with AKI without evidence of parathyroid gland disorders, as well as cases of transient PTH increase during AKI that normalized completely after recovery within 48 h [34,35].

In human CKD, the development of RSHPT is promoted by the interactions between iCa, phosphorus, vitamin D metabolites, PTH and FGF-23. Conversely, in AKI, higher PTH concentrations are primarily driven by hypocalcemia and decreased vitamin D levels, which stimulate the parathyroid glands to secrete PTH [18]. Hypocalcemia in AKI can result from various mechanisms, including reduced renal synthesis of the active form of vitamin D, leading to diminished calcium absorption from the gastrointestinal tract, decreased renal calcium reabsorption and impaired calcium release from bones. Other contributing factors include skeletal resistance to PTH, which limits the hormone’s ability to restore serum calcium levels, upregulation of the calcium-sensing receptors in both the kidneys and parathyroid glands, triggered by proinflammatory cytokines and altering the set point for calcium–PTH feedback regulation and the intracellular calcium accumulation, which occurs in patients with septic AKI [18]. In our cohort of dogs with AKI, median iCa concentration was close to the lower reference limit (1.2 mmol/L), with hypocalcemia detected in a relevant number of dogs, although it is reported that serum calcium concentration is usually within reference range in dogs affected by AKI. Moreover, phosphate concentrations were almost always elevated, consistent with previously reported findings [1]. Therefore, the increase in PTH could represent a rapid compensatory response to both rising phosphate and declining calcium concentrations. PTH plays a critical role in minute-to-minute regulation of blood iCa concentration, acting on intestinal absorption and renal and bone reabsorption [17,36]. Another key function of PTH is to directly promote phosphate excretion by kidneys, reducing its serum concentration. This is consistent with the traditional “trade-off” hypothesis of RSHPT pathophysiology in CKD, which attributes the decreased clearance of phosphate ions and their subsequent rise in plasma as the primary drivers of RSHPT, even in the absence of ionized hypocalcemia [37]. In dogs, as in humans, it can be hypothesized that, during AKI, hypocalcemia and hyperphosphatemia may contribute to the rising in PTH concentrations to restore mineral homeostasis, as indicated by the strong correlation between iCa and P with PTH. Information on vitamin D concentrations, which are often low in dogs affected by CKD [38], could provide additional insights into this process. However, this investigation was not feasible in the present study due to the insufficient amount of remaining serum.

The correlation between iCa and PTH, alongside with the absence of a similar correlation between tCa and PTH, confirms the involvement of PTH in calcium homeostasis and highlights iCa measurement as the most reliable evaluation of biologically active calcium status [39].

Hypercalcemia was detected in dogs with AKI and CKD. While tertiary HPT with an altered set point for PTH secretion may explain hypercalcemia in CKD dogs, its underlying mechanisms in AKI are more challenging to elucidate, in the absence of malignancies or primary hyperparathyroidism. Hypercalcemia should prompt clinicians to consider alternative etiologies but has also to be regarded as a significant contributor to the worsening of AKI. The reduction in glomerular filtration rate caused by afferent arteriolar vasoconstriction, volume depletion induced by the downregulation of the sodium-potassium-chloride cotransporter and consequent natriuresis, and calcium deposits in the kidneys are described as causes for the onset or worsening of the kidney damage [40,41]. In our patients, no evidence of malignancy or of primary hyperparathyroidism was reported at the diagnosis, although volume depletion may have contributed to the condition. This study has some limitations. The first limitation is the retrospective nature of the study itself. These results are made in the absence of some data (e.g., iCa in healthy dogs), which, however, we do not believe significantly affects the validity of our observations. Another limitation could be the inter-laboratory variability, although it is often unavoidable in multicenter studies. However, it is important to note that such variability was only applied to the routine biochemical determinations and not to PTH measurement, which was conducted solely at a single laboratory. Moreover, it is also possible that some patients considered to be grade I AKI could be patients with CKD stage I, given the difficulty in distinguishing between the two interrelated pathologies with a single time point of blood exams.

## 5. Conclusions

This study demonstrates the significant presence of elevated concentrations of PTH in dogs affected by AKI, similar to findings in human AKI. It also establishes a correlation between PTH concentrations and disease severity, as indicated by serum creatinine, IRIS AKI grade, iCa and P concentrations.

The comparison between dogs with AKI and those with CKD did not highlight significant differences in PTH concentrations, suggesting that the increase in PTH may represent an adaptive response to mineral imbalances and renal damage, independent of the time of the onset of renal function decline.

Further studies are needed to elucidate the influence of elevated PTH concentrations on treatment response and prognosis in dogs with AKI and to evaluate the remission of higher PTH concentration following recovery.

## Figures and Tables

**Figure 1 vetsci-12-00131-f001:**
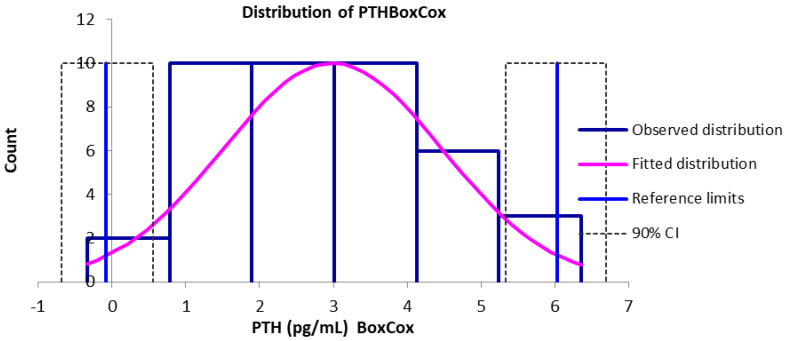
Distribution of PTH in healthy dogs after Box–Cox transformation. PTH = parathyroid hormone.

**Figure 2 vetsci-12-00131-f002:**
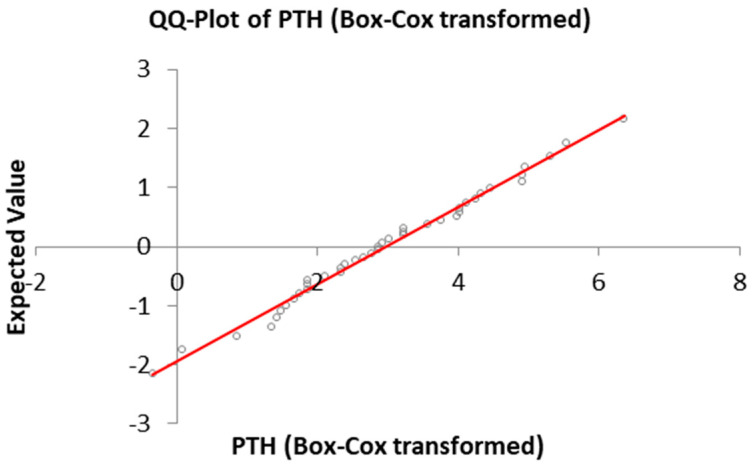
QQ-plot compare the quantiles of the observed variable with those of normal distribution. PTH = parathyroid hormone.

**Figure 3 vetsci-12-00131-f003:**
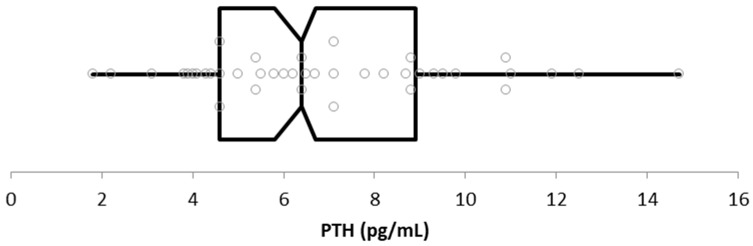
Box plot distribution of PTH concentration in healthy dogs. Boxes represent the interquartile ranges and the inside line the median value. Dots represent the different PTH concentrations with minimum and maximum values.

**Figure 4 vetsci-12-00131-f004:**
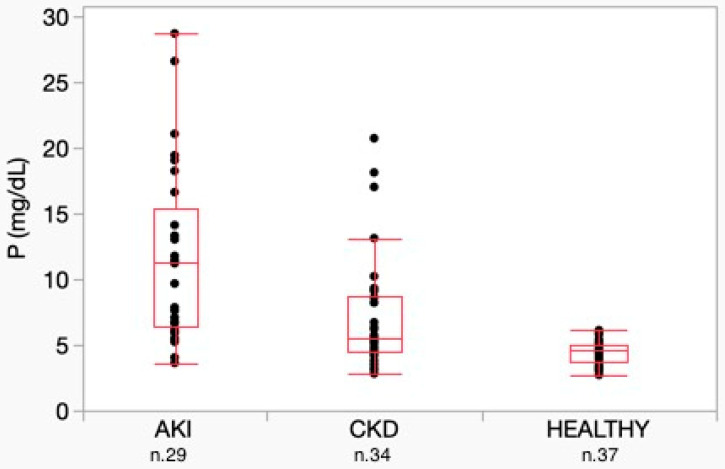
Phosphate concentration between AKI, CKD and healthy dogs. Horizontal lines represent median values. Boxes represent interquartile ranges. Dots represent the different phosphate concentrations. The distribution of analytes was different among groups according to post-hoc Dunn test. AKI = acute kidney injury; CKD = chronic kidney disease; P = phosphate.

**Figure 5 vetsci-12-00131-f005:**
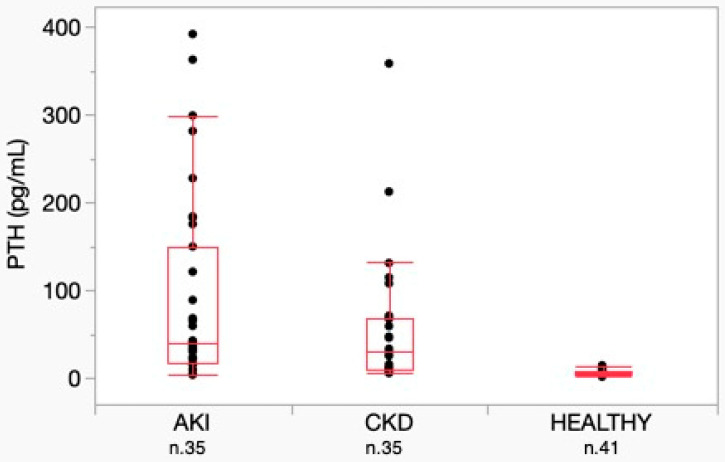
PTH concentration between AKI, CKD and healthy dogs. Horizontal lines represent median values. Boxes represent interquartile ranges. Dots represent the different PTH concentrations. AKI = acute kidney injury; CKD = chronic kidney disease; PTH = parathyroid hormone.

**Figure 6 vetsci-12-00131-f006:**
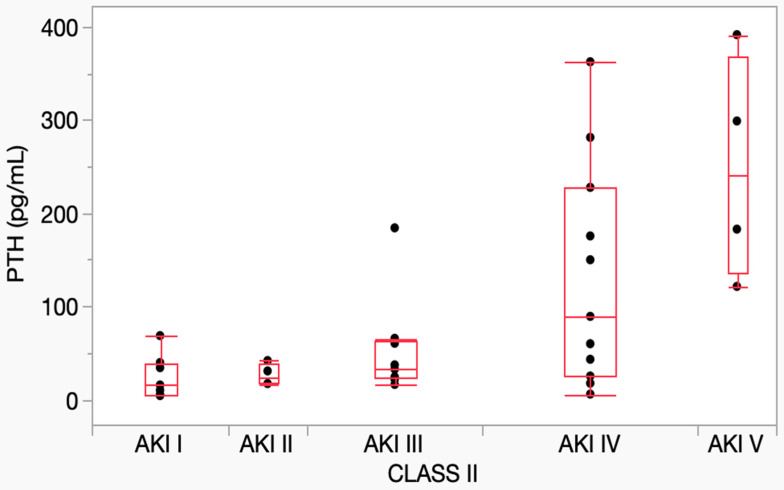
PTH concentration in AKI grades. Horizontal lines represent median values, boxes represent interquartile ranges. Dots represent the single PTH concentration. AKI = acute kidney injury; PTH = parathyroid hormone.

**Table 1 vetsci-12-00131-t001:** Biochemical variables in AKI, CKD and healthy dogs. Data are reported as median and interquartile range; a, b, c: different letters show significant differences between groups. AKI = acute kidney injury; CKD = chronic kidney disease; iCa = ionized calcium; IQR = interquartile range; n.a. = not analyzed; P = phosphate; PTH = parathyroid hormone; sCr = serum creatinine; tCa = total calcium.

Variable (Units)		AKI (n = 36)	CKD (n = 36)	Healthy Dogs (n = 41)	Reference Interval
sCr (mg/dL)	Median	3.7 ^a^	3.4 ^a^	1.09 ^b^	≤1.4 mg/dL
	Min–Max (IQR)	1.38–15.02 (5.4)	0.7–15.6 (3.7)	0.7–1.4 (0.3)	
	Number of samples	n = 35	n = 35	n = 41	
	Number of abnormal	34/35	28/35	1/41	
P (mg/dL)	Median	11.3 ^a^	5.4 ^b^	4.6 ^c^	3.5–6.2 mg/dL
	Min–Max (IQR)	3.6–28.6 (8.9)	2.8–20.7 (4.2)	2.7–6.1 (1.2)	
	Number of samples	n = 29	n = 34	n = 37	
	Number of abnormal	29/29	15/34	7/37	
tCa (mg/dL)	Median	9.7 ^a^	10.5 ^a^	10.4 ^a^	8–12 mg/dL
	Min–Max (IQR)	7.3–17.4 (1.7)	6.4–14.5 (2.6)	6.8–11.6 (1.1)	
	Number of samples	n = 27	n = 33	n = 37	
	Number of abnormal	5/27	11/33	2/37	
iCa (mg/dL)	Median	1.2 ^a^	1.3 ^b^	n.a.	1.2–1.4 mg/dL
	Min–Max (IQR)	0.8–2.1 (0.9)	0.8–1.4 (0.1)		
	Number of samples	n = 28	n = 34		
	Number of abnormal	11/28	8/34		
PTH (pg/mL)	Median	39.4 ^a^	30.4 ^a^	6.4 ^b^	1.8–14.7 pg/mL
	Min–Max (IQR)	3.9–391.6 (132.3)	5.9–358.1 (58.2)	1.8–14.7 (4.3)	
	Number of samples	n = 35	n = 35	n = 41	
	Number of abnormal	31/35	24/35	0/41	

**Table 2 vetsci-12-00131-t002:** PTH concentration in AKI grades and CKD stages. AKI = acute kidney injury; CKD = chronic kidney disease; IQR = interquartile range; IRIS = International Renal Interest Society; PTH = parathyroid hormone.

IRIS Grade/Stage n of AKI/CKD Patients	PTH (pg/mL)	AKI (n = 35)	CKD (n = 35)
1 n of AKI = 7 n of CKD = 7	Median	15.3	16.4
	Min–Max	3.8–68.3	7.1–26.3
	range	64.4	19.2
	IQR	34.4	17.5
2 n of AKI = 4 n of CKD = 9	Median	23.9	16.1
	Min–Max	16.8–41.5	5.9–46.9
	range	24.7	41
	IQR	21.7	13.4
3 n of AKI = 9 n of CKD = 9	Median	32.4	55.7
	Min–Max	15.9–184.1	10.8–107.9
	range	168.2	97.1
	IQR	40	28.4
4 n of AKI = 11 n of CKD = 10	Median	89	118.5
	Min–Max	5.5–362.5	15.9–358.1
	range	357	342.2
	IQR	202.7	108.5
5 n of AKI = 4	Median	240.8	/
	Min–Max	121.2–391.6	
	range	270.4	
	IQR	231.8	

## Data Availability

Data are available in the database of the Veterinary Teaching Hospitals.

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
