# Peer review of "Parathyroid Hormone Concentration in Dogs Affected by Acute Kidney Injury Compared with Healthy and Chronic Kidney Disease"

_vetsci, 2025, doi:10.3390/vetsci12020131_

Round 1
Reviewer 1 Report
Comments and Suggestions for Authors
The experimental design used was a retrospective study. Some information needs to be clarified for me. Do the places where the dogs were evaluated have a similar protocol for assessing patients with kidney disease? Is parathyroid hormone measured routinely in patients with kidney disease?
I also have some questions about the group of healthy dogs. What complaints were the animals considered healthy taken to the clinic for? Were these dogs suspected of having kidney disease, so they underwent the same laboratory tests?
L.76-83: Make it clear that information refers to humans.
L.186: The item name must be corrected.
Author Response
Comments 1: The experimental design used was a retrospective study. Some information needs to be clarified for me. Do the places where the dogs were evaluated have a similar protocol for assessing patients with kidney disease? Is parathyroid hormone measured routinely in patients with kidney disease?
Response 1: The patients were evaluated in two different Institutions with a similar approach regarding diagnosis and treatment of renal diseases based on IRIS guidelines. The criteria for diagnosis are reported in the “materials and methods” paragraph. PTH was not routinely measured but measured on leftover frozen serum within 6 months from the withdrawal. The information has been added to the main text. As reported in the paper about the validation of the analytical method cited in the “materials and methods” paragraph, it is demonstrated that PTH is stable in frozen serum until 6 months.
Comments 2: I also have some questions about the group of healthy dogs. What complaints were the animals considered healthy taken to the clinic for? Were these dogs suspected of having kidney disease, so they underwent the same laboratory tests?
Response 2: Thanks for your comment. The group of healthy dogs included patients referred for routinely check-up or blood test before elective surgery such as neutering. PTH was measured on leftover serum, if available. The information has been added to the main text.
Comment 3: L.76-83: Make it clear that information refers to humans.
Response 3: Thanks for the suggestion. The sentence has been rephrased for better comprehension. The main drivers for HPT are hypocalcemia and secondarily hyperphosphatemia and decreased calcitriol
Comment 4: L.186: The item name must be corrected.
Response 4: Thank you for the suggestion. The title has been modified as requested

Reviewer 2 Report
Comments and Suggestions for Authors
Dear Authors,
The Manuscript had a promising aim to test in the acute kidney injury (AKI) settings the relevance of hypoparathyroidism, a feature commonly appearing in chronic kidney disease (CKD). Accordingly, the results showed the potential in pathophysiological and clinical pathology context. Before the acceptance for publication, minor corrections and additional explanations seem necessary, as listed below.
Lines:
Introduction
- 41‒2: To enhance interpretation, please additionally explain differentiating challenges because AKI and CKD differ in symptomatology dynamics (as detailed in the first two paragraphs), thus not necessarily depending on creatinine concentration. The reference(s) should support the explanation.
- 76‒83: Please clarify that the remarks refer to human medicine.
Materials and methods
- 106: Which reference did you follow in establishing USG criteria?
- 115‒24: Please include the details about the tubes used for blood collection, from which serum was collected after the centrifugation, and an explanation for which tests you used each of the mentioned instruments (BT 3500, AU480, phOX Ultra Analyzer, and the ABL825).
Results
In general, please consider additional efforts to avoid repetition because currently, in several instances, the same data are presented in the Tables, textually, and in the graphs.
- 141‒62: The percentages seem unnecessary, considering the total number of dogs per group. Please provide the P-values for age comparison between the groups.
A 95% confidence interval should accompany The correlation coefficients. Also, please consider correlation assessment in healthy dogs.
- 251‒3: Please clarify that these data refer to the AKI group.
Discussion
- Please consider additional efforts to avoid repeating the results in this section.
- The remarks about study limitations are lacking.
Technical suggestions
- In describing the differences, please avoid the term statistically significant. The significance obtained in the corresponding statistical test was the evidence of the difference.
- The P-values format should be uniform, containing three decimals.
Comments on the Quality of English Language
Please consider additional language editing to improve the overall Manuscripts
Author Response
Comments 1: 41‒2: To enhance interpretation, please additionally explain differentiating challenges because AKI and CKD differ in symptomatology dynamics (as detailed in the first two paragraphs), thus not necessarily depending on creatinine concentration. The reference(s) should support the explanation.
Response 1: We have revised the sentence, trying to emphasize the suggested points. However, we are reluctant to expand this section further, as the subsequent paragraphs already include the limitations of serum creatinine and the need for additional biomarkers, with the appropriate references. This decision also considers the suggestion from another reviewer to shorten the introduction.
Comments 2: 76‒83: Please clarify that the remarks refer to human medicine.
Response 2: In the sentence we have clarified that the observations were conducted on human patients
Materials and methods
Comments 3: 106: Which reference did you follow in establishing USG criteria?
Response 3: Thanks for the question. The reference is always IRIS guidelines. We have moved the reference to the end of the sentence. Recently, the education page has been updated and can now be found at the following address: https://www.iris-kidney.com/urine-specific-gravity. If possible, we would prefer to include a single reference, specifically the website's homepage, to avoid redundant references.
Results
In general, please consider additional efforts to avoid repetition because currently, in several instances, the same data are presented in the Tables, textually, and in the graphs.
Comments 4: 115‒24: Please include the details about the tubes used for blood collection, from which serum was collected after the centrifugation, and an explanation for which tests you used each of the mentioned instruments (BT 3500, AU480, phOX Ultra Analyzer, and the ABL825).
Response 4: Thanks for the suggestion. We have added most of the information requested. The two clinical biochemistry analyzers were located at the University of Milan (BT 3500, Biotecnica Instruments) and at the University of Bologna (Beckman Coulter) and were used for routine testing of hospitalized patients. The analyses included a biochemical and electrolyte profile. Please let us know if you would like a detailed list of the tests performed or if the clarifications provided in the text are sufficient.
Comments 5: 141‒62: The percentages seem unnecessary, considering the total number of dogs per group. Please provide the P-values for age comparison between the groups.
Response 5: We have removed the percentage as suggested. We have added the information requested
Comments 6: A 95% confidence interval should accompany the correlation coefficients. Also, please consider correlation assessment in healthy dogs.
Response 6: We have added the data requested.
Comments 7: 251‒3: Please clarify that these data refer to the AKI group.
Response 7: Thank you for the suggestion, we clarified the sentence.
Discussion
Comments 8: Please consider additional efforts to avoid repeating the results in this section.
Response 8: We have eliminated the repetition of the results as requested.
Comments 9: The remarks about study limitations are lacking.
Response 9: Thank you for the reminder: we have added remarks about limitations.
Technical suggestions
Comments 10: In describing the differences, please avoid the term statistically significant. The significance obtained in the corresponding statistical test was evidence of the difference.
Response 10: We have removed or replaced the term statistically.
Comments 11: The P-values format should be uniform, containing three decimals.
Response 11: We have standardized the decimals.

Reviewer 3 Report
Comments and Suggestions for Authors
This manuscript describes PTH and compares PTH serum concentrations in dogs affected by AKI, CKD and a healthy control group. The novelty of the manuscript is finding elevated PTH concentrations in dogs affected by AKI, as this has not previously been described in this species.
General concept comments:
· A hypothesis is lacking
· In my opinion the alignment of the different parts of the manuscript could be improved.
o The aim is not covering all the results.
o The introduction has a large part considering the differentiation between CKD and AKI, leading to expectations for the rest of the manuscript, but this is not covered by the aim, and only briefly touched upon in the discussion. Is this part of the introduction relevant?
· As the authors report in the discussion, it might be discussed whether the increased serum PTH changes observed is in fact hyperparathyroidism. It might be better to rephrase the aim of the study, using parathyroid hormone concentration (as done in the title and mostly in the conclusion), instead of HPT.
· Both the terms HPT and RSHPT are used, and the distinction is not always clear to me.
· As stated by the authors in the introduction it might me difficult to distinguish CKD from AKI, and this might also be a possible weakness in the present study and should be addressed in the discussion
More specific comments:
Introduction
· Line 43: “sCr is insensitive to acute changes in renal function”, please add reference for this statement or delete
· Lines 76-83: I would suggest adding the species that have been studied in the refences you report in this section, as this is not data from dogs I would guess.
· Line 295; ref 18 – please add species that was studied to help the reader understand
Materials and methods:
· Time from sampling until PTH analyses should be stated
· For the healthy control group, you describe the exclusion criteria, but not the inclusion criteria. Were clinical pathological data available (blood and urine) for all controls?
·
Results:
· I would suggest to use a table to describe the populations (section 3.1)
· Hydration status and fluid treatment (and possible other treatments?) might affect the results in th case groups, please address this by adding data. If not available, discuss this weakness of the study.
· Sections 3.2 and 3.3 have the same title.
· Line 165 – what do you mean by above described?
· Parts of the same data is reported in Table 1 and figure 4 and 5. Please consider whether you need both.
· Lines 199 – 213. In addition to comparing the study groups you compare to refence intervals. Is it useful to do both? If you want to report how many values were outside reference intervals, consider reporting in a table for all analytes and all groups. You should also report the RI in the same table.
· Figure 4 and 5: add N
· Table 2 – Add N for the different stages/grades
Comments on the Quality of English Language
There are some flaws, should be proof read by a fluent speaking
Author Response
Comments 1: A hypothesis is lacking
Response 1: We have modified the introduction section specifying the aim of the study
Comments 2: In my opinion the alignment of the different parts of the manuscript could be improved
Response 2: The authors followed journal’s guidelines regarding paragraphs alignment. The authors also asked which paragraphs should be better aligned.
Comments 3: The aim is not covering all the results.
Response 3: In the introduction paragraph, the aim was specified as requested.
Comments 4: The introduction has a large part considering the differentiation between CKD and AKI, leading to expectations for the rest of the manuscript, but this is not covered by the aim, and only briefly touched upon in the discussion. Is this part of the introduction relevant?
Response 4: In the introduction paragraph, the aim has been modified hypothesizing different PTH concentrations between AKI and CKD patients as a differentiating tool between these patients.
Comments 5: As the authors report in the discussion, it might be discussed whether the increased serum PTH changes observed is in fact hyperparathyroidism. It might be better to rephrase the aim of the study, using parathyroid hormone concentration (as done in the title and mostly in the conclusion), instead of HPT.
Response 5: The authors have modified the aim of the study using PTH concentration instead of HPT.
Comments 6: Both the terms HPT and RSHPT are used, and the distinction is not always clear to me
Response 6: The authors have modified the terms used in the manuscript distinguishing RSPHT in CKD patients and higher PTH concentration during AKI.
Comments 7: as stated by the authors in the introduction it might be difficult to distinguish CKD from AKI, and this might also be a possible weakness in the present study and should be addressed in the discussion.
Response 7: The authors have added this study limitation as suggested.
Comments 8: Line 43: “sCr is insensitive to acute changes in renal function”, please add reference for this statement or delete.
Response 8: The authors have specified the reference at the end of the statement.
Comments 9: Lines 76-83: I would suggest adding the species that have been studied in the refences you report in this section, as this is not data from dogs I would guess.
Response 9: The authors have specified in the text that the cited studies considered human patients, as requested.
Comments 10: Line 295; ref 18 – please add species that was studied to help the reader understand.
Response 10: The authors have modified the sentence as requested.
Comments 11: Time from sampling until PTH analyses should be stated.
Response 11: The authors have modified the sentence with the time from sampling till PTH evaluation.
Comments 12: For the healthy control group, you describe the exclusion criteria, but not the inclusion criteria. Were clinical pathological data available (blood and urine) for all controls?
Response 12: Clinicopathological data for the healthy groups include biochemical parameters together with urine exam with proteinuria quantification. Specifically, biochemical data included serum creatinine, total calcium, and phosphate. Together with the biochemical exams, also complete blood count analysis was made.
Comments 13: I would suggest to use a table to describe the populations (section 3.1)
Response 13: Thanks for the suggestion. The authors prefer to maintain the part of population information written, to better describe patients’ characteristics.
Comments 14: Hydration status and fluid treatment (and possible other treatments?) might affect the results in th case groups, please address this by adding data. If not available, discuss this weakness of the study.
Response 14: For the three study groups, blood samples were collected upon admission, as specified in the text.
Comments 15: Sections 3.2 and 3.3 have the same title
Response 15: The authors have modified the 3.3 section as requested.
Comments 16: Line 165 – what do you mean by above described?
Response 16: The authors have modified the phrase as requested.
Comments 17: Parts of the same data are reported in Table 1 and figure 4 and 5. Please consider whether you need both.
Response 17: Thanks for the suggestion. Table 1 consisted of written laboratory data, whereas figures depict more intuitively PTH and phosphate concentrations among groups.
Comments 18: Lines 199 – 213. In addition to comparing the study groups you compare to refence intervals. Is it useful to do both? If you want to report how many values were outside reference intervals, consider reporting in a table for all analytes and all groups. You should also report the RI in the same table.
Response 18: The authors have modified Table 1 with the Reference Interval column and the number of abnormal values.
Comments 19: Figure 4 and 5: add N
Response 19: The authors have added N as requested
Comments 20: Table 2 – Add N for the different stages/grades
Response 20: The authors have added N as requested
